# Reconstruction of Urban Rainfall Measurements to Estimate the Spatiotemporal Variability of Extreme Rainfall

**Risma Joseph, P. P. Mujumdar and Rajarshi Das Bhowmik \***

Interdisciplinary Centre for Water Research, Indian Institute of Science, Bangalore 560012, India
\* Correspondence: rajarshidb@iisc.ac.in; Tel.: +91-080-2293-3224

**Abstract:** In recent decades, the impact of climate change on urban flooding has increased, along with an increase in urban population and urban areas. Hence, historical design storms require revisions based on robust intensity–duration–frequency (IDF) relationships. To this end, the development of an urban rain-gauge network is essential to yield the spatiotemporal attributes of rainfall. The present study addresses two objectives: (a) to reconstruct sub-daily rainfall time series for the historical period over an urban gauge network, and (b) to investigate the spatiotemporal variation in extreme rainfall distribution within a city. This study considers Bangalore, India, where rainfall has been historically monitored by two stations but a dense gauge network has recently been developed. The study applies random forest regression for rainfall reconstruction, finding that the performance of the model is better when the predictand and predictor stations are near to one another. Robust IDF relationships confirm significant spatial variations in extreme rainfall distribution at the station and the city-region levels. The areal reduction factor (ARF) is also estimated in order to understand the likely impact of the reconstructed time series on hydrological modeling. A significant decrease in the ARF is observed as the area grows beyond 450 km$^2$, indicating a substantial reduction in the volume of the design floods.

**Keywords:** rainfall reconstruction; extreme rainfall; random forest; uncertainty; IDF; areal reduction factor

## 1. Introduction

Urban areas have been experiencing frequent heavy flooding resulting from extreme rainfall events attributed to climate change [1,2]. Rapid urbanization has been affecting the natural land cover of regions, creating urban heat islands that intensify precipitation through additional instability and greater moisture transport [3] (Huang et al., 2022). Urban structures and car emissions are also worsening urban flood conditions [4]. In the recent past, several metro cities in India have witnessed high-intensity rainfall and subsequent floods as surface runoff exceeds the drainage capacity [5–7]. Therefore, urban flood prediction at different lead times has attracted significant attention from researchers. However, reliable flood prediction critically depends on the model forcing, since the forcing can impart significant uncertainty to the prediction [8]. Prediction models are sensitive to meteorological forcing, such as precipitation and temperature, since the forcing may undergo significant spatiotemporal variations. The present study examines the spatiotemporal variation in extreme rainfall for an Indian city (Bangalore) to understand the importance of reconstructing urban rainfall measurements.

Traditionally, rainfall has been measured using rain gauges, whose estimates are considered to represent the direct point measurement of precipitation. At a regional and/or urban scale, due to the sparse distribution of rain gauges, interpolation techniques are adopted to estimate the rainfall at locations that are not covered by a rain gauge [9–11]. Over the last few decades, satellite- and model-based finer-resolution estimates of rainfall have been gaining popularity; however, the point estimates of rainfall from gauge stations

remain useful as ground truths. The authors note that the spatial distribution of rain-gauge stations is critical because it depends primarily on the spatial variability of rainfall and the funds available for the development and maintenance of the stations. Several studies have been conducted to design optimal rain-gauge networks accounting for the spatial distribution of rainfall [12]. Traditionally, only a few stations are built within a city's limits, assuming spatial homogeneity in rainfall over the city. It should be noted that the density of rain-gauge networks within a city might be even higher than in rural areas [13]. Several previous studies have investigated the importance of developing distributed rain-gauge networks for urban catchments [13,14]. Furthermore, with the growing impact of human-induced climate change and land-use changes, urban rainfall characteristics are also expected to undergo significant changes. The majority of the previous studies related to urban rain-gauge networks consider the stationary statistical attributes of extreme rainfall events. To the best of our knowledge, very few have investigated the importance of a distributed rain-gauge network in yielding non-stationarity in extreme rainfall distribution.

Previous studies have reported that spatiotemporal changes in urban rainfall attributes are significantly different from those in non-urban rainfall attributes [4,15]. The impact of urbanization is not only limited to the enhancement of mean precipitation—it can also impact the mesoscale extreme rainfall [16,17]. Common practices to understand the spatial variability of rainfall within a city include (i) employing a numerical weather prediction model to simulate the event-scale rainfall estimates [18], and (ii) considering rain-gauge data or satellite and radar estimates [19]. One potential limitation of numerical weather prediction models is that their outputs yield substantial uncertainty arising from boundary conditions, initial values, and model structures [20,21]. In contrast, satellite and radar estimates require efficient post-processing based on the ground truth, signifying the importance of developing and maintaining an urban rain-gauge network. Considering the limitations and requirements of urban rain-gauge networks, a potential alternative is to reconstruct the historical rainfall time series from the recently developed dense urban rain-gauge network. The reconstruction of daily–sub-daily rainfall time series is a well-known concept that employs interpolation techniques, regression-based approaches, clustering algorithms, tree-ring chronology, and stable-isotope-based techniques, and serves a promising alternative to a long-running urban rain-gauge network [22–25]. Considering these factors, the present study addresses the following objectives:

1. To reconstruct the sub-daily rainfall time series for an urban rain-gauge network using a machine learning algorithm.
2. To investigate the spatiotemporal changes in extreme rainfall for Bangalore, India, with an additional focus on the intracity variations.

The city of Bangalore has experienced rapid urbanization in recent decades. It has the largest carbon footprint among all Indian cities. Traditionally, rainfall in Bangalore has been monitored by two meteorological observation stations maintained by the India Meteorological Department (IMD)—a federal organization. In a previous study, Rupa and Mujumdar [26] investigated the spatiotemporal changes in extreme rainfall over Bangalore using the data observed at 17 IMD stations across urban and suburban Bangalore. In recent years, considering the growing concern of urban flooding, a local governmental agency—the Karnataka State Natural Disaster Management Centre (KSNDMC)—installed a dense network of rain-gauge stations across the city. The present study developed a random forest model between rainfall data from the IMD station (as predictors) and rainfall observations from the KSNDMC stations (as predictands) for a common time period to address the first objective. Furthermore, to address the second objective, we performed two tasks: (a) identification of non-stationarity in the annual maximum rainfall (AMR) series from the reconstructed data using the ADF test, and (b) the development of robust IDF relationships for the stationary and non-stationary AMR series. Finally, we calculated an areal reduction factor (ARF) to estimate the ratio between the point and the areal average rainfall estimates. The ARF indicates the likely impact of design flood computation if the point rainfall is used in the computation as areal average rain. A dense rain-gauge network

assists the modelers in developing a relationship between the urban area and the ARF, which can be applied to the point rainfall estimate prior to design flood computation. The following section discusses the data used in this study and provides a brief description of the study area. This is followed by an explanation of the methodology used in the study. The results of the study are presented in Section 4, which is followed by a summary and discussion.

## 2. Data and Study Area

### 2.1. IMD and KSNDMC Station Data

The hourly rainfall data from the IMD stations in Bangalore—IMD 43,295 and IMD 43,296—are available from 1969 to 2019. The KSNDMC provides real-time weather-related information, forecasts, early warnings, and advisories for the management of natural disasters in the state. The KSNDMC has installed over 6000 telemetric rain gauges (TRGs) and more than 750 telemetric weather stations (TWSs) across the entire state to transmit data every 15 min. The hourly rainfall data from 51 KSNDMC stations over Bangalore were considered from the years 2010 to 2019. The locations of the IMD and KSNDMC stations are shown in Figure 1a. For demonstration, the rainfall time series for the two IMD stations and one representative KSNDMC station (TRG2309) are shown in Figure 1b,c, respectively. Additional details related to the IMD stations, originally mentioned by Rupa and Mujumdar (2018), are provided in Table 1.

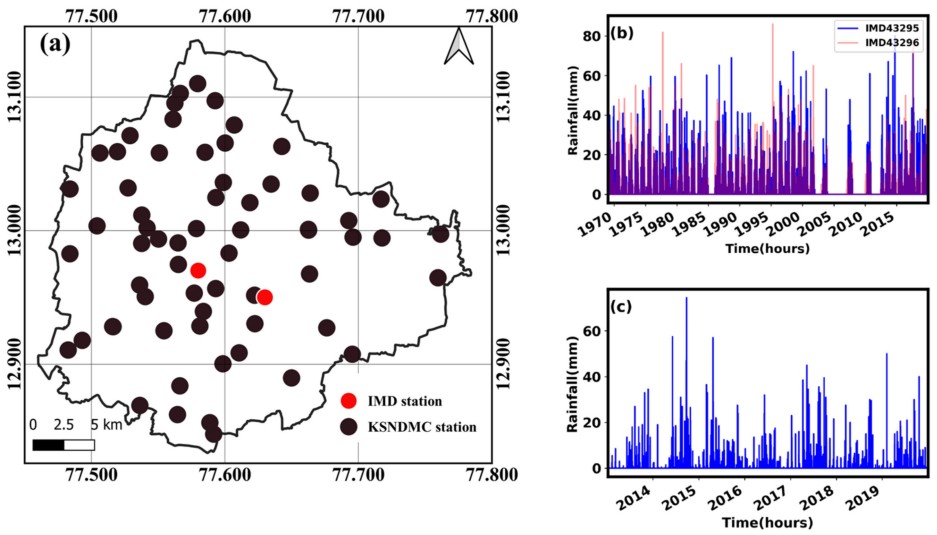

**Figure 1.** (**a**) Locations of the IMD and KSNDMC stations. (**b**) Hourly rainfall time series for the two IMD stations. (**c**) Hourly time series for a representative KSNDMC station (TRG 2309).

**Table 1.** Details related to the two IMD stations located in Bangalore.

| SI. No. | Station Name | Index No. | Latitude | Longitude | Elevation |
|---------|--------------|-----------|----------|-----------|-----------|
| 1 | City | 43295 | 12.97° N | 77.58° E | 911 m |
| 2 | HAL | 43296 | 12.95° N | 77.63° E | 899 m |

### 2.2. Study Area: Bangalore

Bangalore, lying between 77.5° E–77.8.0° E and 12.8° N–13.2° N, with a population of 9 million (as per the 2011 census), was considered for this study (see Supplementary Figure S1). The city is located in the southern part of India and experiences a tropical savanna with dry winters as per the Köppen climate classification [27]. The elevation varies between 620 m and 1082 m over an area of approximately 740 km². As per Bhuvan—India's geo-platform—almost 37% of Bangalore is built-up urban land, followed by 4% built-up

mining areas and 2% agricultural plantations. Bangalore contains multiple lakes, which are major sources of fresh water for the city. The city receives precipitation from the southwest (during June–September) and northeast (during October–December) monsoons, with an average annual precipitation of around 974 mm. Summer (March–May) precipitation, influenced by localized convective heat transfer, frequently leads to intense flooding [26]. Additionally, the city experiences average maximum and minimum temperatures of 36° and 14 °C, observed during April and January, respectively. The relative humidity in Bangalore varies between 35 and 80% [28].

## 3. Methodology

The present study is divided into three major tasks: (i) reconstruction of the hourly time series for 51 KSNDMC stations for the period 1969–2019, (ii) spatiotemporal analysis of the reconstructed rainfall series, and (iii) computation of the areal reduction factors (ARFs) for different storm durations. A flowchart of the tasks is provided in Figure 2.

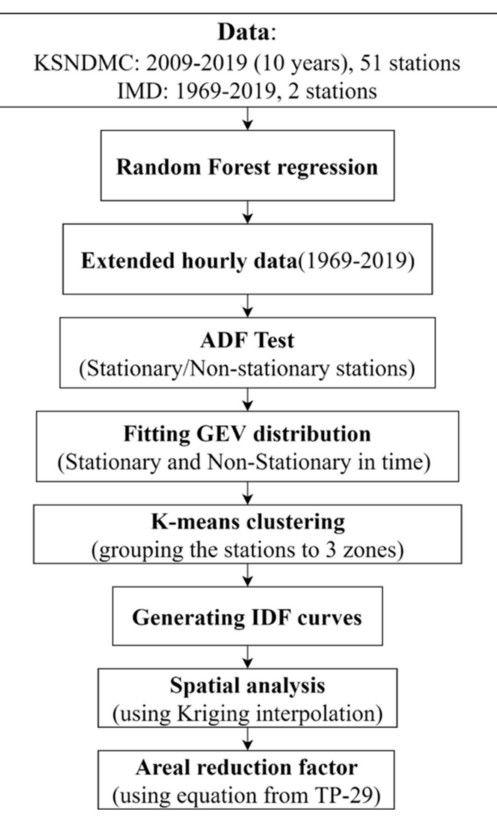

**Figure 2.** Schematic diagram showing the primary tasks of this study.

### 3.1. Rainfall Reconstruction Using Random Forest Regression

To address the first task, we developed a random forest regression model between the IMD station rainfall data (as predictors) and the rainfall time series from a KSNDMC station (as predictands) for the period 2010–2019. We employed a two-fold model calibration and validation procedure to ensure the absence of data overfitting. Two-thirds of the data were used for calibration, while the rest were used for validation. The random forest model is a tree-based model where multiple tree blocks form a forest, where the algorithm provides an average of the predictions over individual trees. A bootstrap sample was created by uniformly resampling the input dataset by replacement. Following this, decision trees were built using the resampled datasets—a process also known as bootstrap aggregation or bagging [29]. Additionally, a manual trial-and-error procedure was performed to decide the number of trees, which was allowed to vary between stations. The model can be run multiple times to develop a model parametric uncertainty estimate, which is not very

high for the current problem (results not shown). Further details related to the random forest model are provided in Appendix A. The model's performance was evaluated on the basis of the coefficient of determination. A coefficient of determination value close to 1 ensures a perfect model fit, while a value above 0.5 is traditionally considered acceptable in hydrometeorological modeling [30]. Prior to the model fitting, we applied a Box–Cox transformation on the predictors and predictands. Furthermore, separate random forest models were developed for the different months to account for seasonality—particularly for the monsoon months. Finally, following calibration and validation, the model was applied to the historical rainfall measurements (1969–2019) from the IMD stations to reconstruct the hourly rainfall time series for the KSNDMC stations. We should note that the predictors to the model at any given time are the observations from the two IMD stations. Hence, the predictors do not vary for the KSNDMC stations; only the functional relationship between the predictands and predictors varies between the KSNDMC stations.

### 3.2. Spatiotemporal Analysis of the Reconstructed Rainfall

For the second task, we first investigated the annual maximum rainfall (AMR) series for non-stationarity. To achieve this, a block maxima approach was applied to the reconstructed hourly rainfall time series to determine the AMR at each KSNDMC station. An augmented Dickey–Fuller (ADF) test was used to identify the KSNDMC stations experiencing non-stationarity in their AMR. The ADF test examines the null hypothesis that a unit root exists while considering a higher-order autoregressive process. The details of the ADF test are provided in Appendix B. Subsequently, stationary and/or non-stationary generalized extreme value (GEV) distributions were fitted to the AMR series over each KSNDMC station. A stationary GEV distribution is a three-parameter function where the three parameters (i.e., location, scale, and shape) do not vary with time (Equation (1)). Additionally, the present study considers a non-stationary GEV distribution where the location parameter is considered to be linearly dependent on time (Equation (2)). It should be noted that non-stationarity in the AMR series can be modeled with multiple functional relationships that associate the GEV parameters with the time variable. However, the present study considers a simple non-stationary GEV model to understand the spatiotemporal variation in IDF relationships.

$$X_t \sim GEV\left(\mu,\ \sigma,\ \xi\right) \tag{1}$$

$$X_t \sim GEV\left(\mu(t) = \beta0 + \beta1t\ ,\ \sigma,\ \xi\right) \tag{2}$$

In Equations (1) and (2), $\mu$, $\sigma$, and $\xi$ are the location, scale, and shape parameters, respectively, whereas $\beta0$ and $\beta1$ are linear model parameters related to the location parameter. The first step towards developing an IDF relationship—whether for a gauge station or a region—is to obtain and process (if required) the GEV parameters. The following equation is considered to estimate the intensity corresponding to a frequency $r$:

$$\hat{z}_r = \hat{\mu} + \frac{\hat{\sigma}}{\hat{\xi}}\left[\left(-\log\left(1 - r^{-1}\right)\right)^{-\hat{\xi}} - 1\right] \tag{3}$$

If a KSNDMC station is experiencing stationary AMR series, Equation (3) can be directly applied to develop an IDF relationship. However, if the AMR series for a KSNDMC station is experiencing non-stationarity, the GEV parameters require further processing. To account for the non-stationarity in an IDF relationship, we applied the method of Feitoza Silva et al. [31] by considering the 95th and 5th percentile values of the location parameter for a given KSNDMC station. To estimate the spatial variation in the IDF relationships across the city, rainfall intensities—given the storm duration and storm design frequency across the KSNDMC stations—were spatially interpolated using kriging interpolation at a spatial resolution of 50 m. Additionally, developing an IDF relationship over a region rather than for gauge stations is common practice. Hence, this study considers a spatial median of the GEV parameter values for the gauge stations within a region to estimate the rainfall intensities, as shown in Equation (3). For example, if the gauge stations within a region are

non-stationary, the spatial median, spatial 75th percentile, and spatial 25th percentile values of the 5th and 95th percentiles are considered during the calculation of rainfall intensity to address the spatiotemporal uncertainty. The current approach combines both the temporal and spatial variations of the GEV parameter values to yield critical information relating to a region. The city was grouped into three regions (later referred to as city-regions) by a k-means clustering method based on the latitude, longitude, and elevation [32]. The details of the k-means clustering are provided in Appendix C.

### 3.3. Computation of ARF

In the final task, we estimated an areal reduction factor (ARF) to understand the importance of the reconstructed rainfall series in hydrological modeling. When applied to the point rainfall, an ARF provides the areal rainfall for a given intensity and a given storm duration; hence, it ranges between 0 and 1 [33]. While several approaches have been considered in previous studies to estimate the ARF, the present study applies a common approach that was originally suggested by the US Weather Service in Technical Paper 29 [34]. We considered four durations (15 min, 30 min, 1 h, and 2 h) for the ARF calculation. Further details related to the ARF estimation are provided in Appendix D.

### 4. Results

### 4.1. Reconstruction of Historic Rainfall

Results related to the cross-validation of the random forest model are presented in Figure 3. The coefficient of determination (R2) values for the KSNDMC stations are marked in various colors. For reference, the locations of the IMD stations are marked in red. We found that KSNDMC stations closer to an IMD station typically show a higher coefficient of determination (R2) value as compared to KSNDMC stations that are far from IMD stations. In particular, the KSNDMC stations near IMD 43,295 exhibited higher R2 values compared to the KSNDMC stations near IMD 43,296. This observation may be related to the greater accuracy of IMD 43,295 than of IMD 43,296 in monitoring sub-hourly rainfall. We found that the performance of the random forest regression was better during February and December than in the other months, because the R2 for these two months was greater than 0.6. However, the performance of the model was relatively poorer during April, July, and September, as several KSNDMC stations experienced an R2 of less than 0.6 during these months. Overall, we found that the random forest regression performs satisfactorily during the validation period, as it captures the error variance efficiently. Hence, this model can be deployed for the historical reconstruction of hourly rainfall series over KSNDMC stations. The representative results of this reconstruction for three KSNDMC stations (2306, 2309, and 2318) for the year 2019 are presented along with the observed rainfall in Figure 4. The present study did not plot a reconstructed time series for earlier periods, since an observed time series was not available over any KSNDMC station. The results indicate that although the reconstructed time series exhibits patterns that are similar to the observations, it typically underestimates the high extremes, which might result in an overall underestimation of the rainfall climatology. We found that the observed extremes and the predicted extremes had a Pearson's correlation coefficient of around 0.7 for 30 stations (see Table S1 for additional details). Nevertheless, it should be noted that machine learning models typically underestimate the extremes; hence, the final outcome should not be influenced by the inefficiency of random forest models.

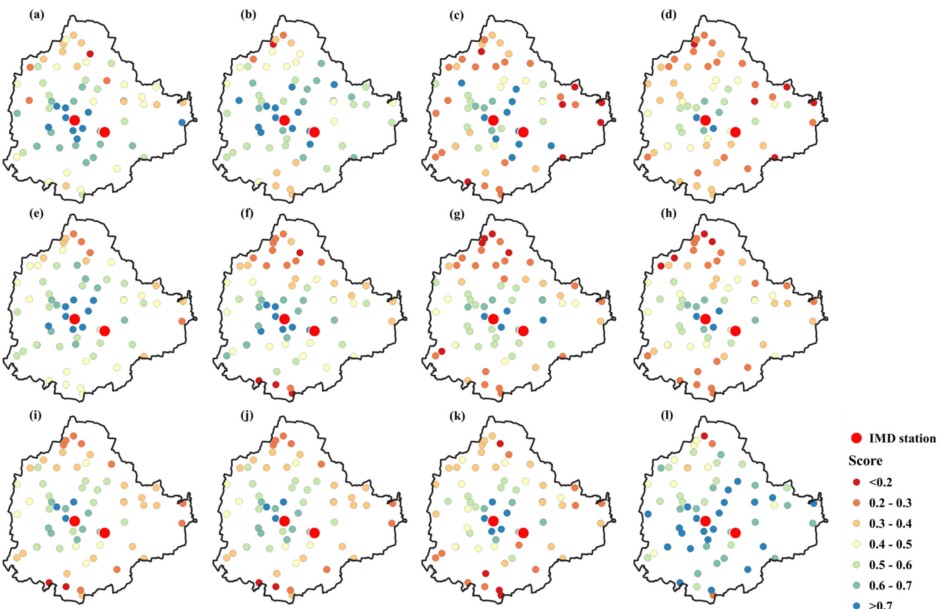

**Figure 3.** Performance of the random forest model shown for 12 months ((**a**–**l**) January–December). The coefficient of determination values for the KSNDMC stations are marked in various colors. Additionally, the locations of the two IMD stations are marked in red.

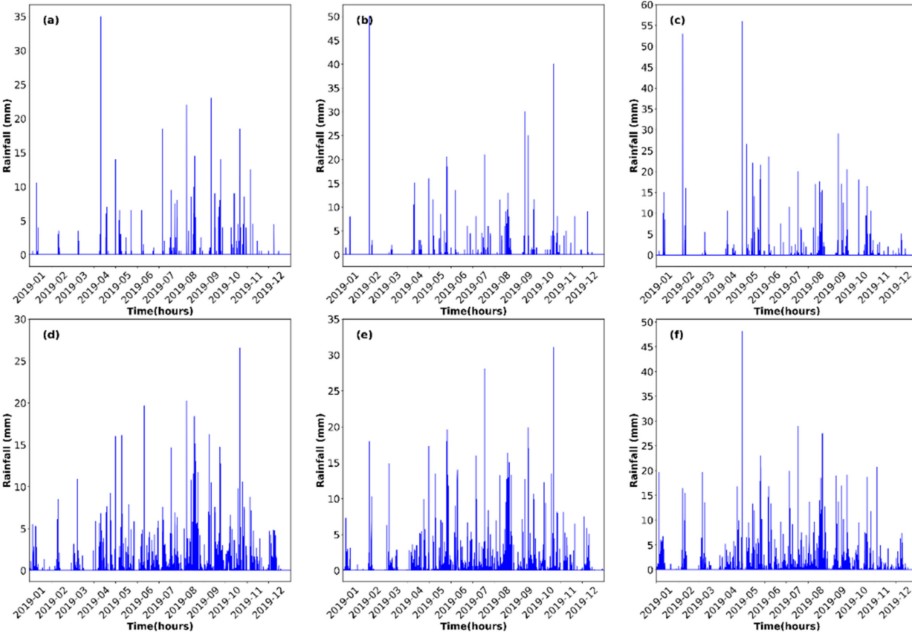

**Figure 4.** Observed (**a**–**c**) and reconstructed (**d**–**f**) time series of hourly rainfall for three representative KSNDMC stations (station numbers: 2306, 2309, and 2318) for the year 2019.

### 4.2. Spatiotemporal Analysis of the Extremes

Results related to the non-stationary analysis of the AMR are presented in Figure 5. Figure 5a depicts the KSNDMC stations that experienced non-stationarity in their AMR following an ADF test. The AMR series for the KSNDMC stations were developed using the reconstructed time series. A stationary GEV model test was fitted over the KSNDMC stations that rejected the null hypothesis of the ADF. The stationary GEV model parameters are shown in Figure 5b. The interquartile ranges in the boxplots for Figure 5b,c represent the spatial variation in the stationary and non-stationary GEV parameters across the KSNDMC stations, respectively. We found that 57% of the KSNDMC stations experienced stationarity

in their AMR series, as opposed to the 43% that experienced non-stationarity. As expected, the scale and shape parameter values remained similar between the stationary and non-stationary GEV models. The location parameter in a non-stationary GEV model is a function of time. However, we found that the trend component ($\beta1$) was small in the non-stationary model. Therefore, the absolute mean location parameter of the stationary model remained the same as the intercept term ($\beta0$) of the non-stationary model. As mentioned in Section 3, this study adopted a slightly different approach in constructing IDF relationships between the stationary and non-stationary KSNDMC stations.

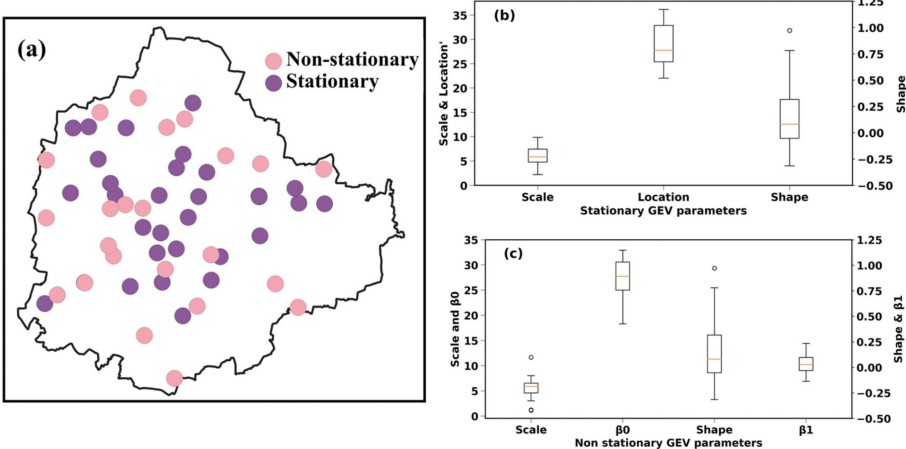

**Figure 5.** (**a**) KSNDMC stations that experienced non-stationarity in their annual maximum rainfall series following an ADF test. (**b**,**c**) The parameters of the stationary and non-stationary GEV models, respectively.

Spatial plots for the rainfall intensity values, given the storm duration and storm frequency, are presented in Figure 6. To avoid further complications in the computation, we considered only the stationary KSNDMC stations. Rainfall intensity values over the KSNDMC stations were spatially interpolated over Bangalore using the kriging approach. In Figure 6, the rows represent three durations—1, 3, and 5 h—while the columns represent three return periods: 10, 25, and 50 years. The results show that the rainfall intensity is highest at the center of the city for any storm duration and frequency. The spatial heterogeneity in rainfall intensity is substantially higher for shorter storms as compared to the longer storms, which is potentially linked to the influence of scattered thunderstorms on the short-duration storms. Longer storms in the city typically occur as a result of the monsoonal and post-monsoonal deep depressions covering the entire city which, in turn, cause spatial homogeneity in the rainfall intensity. The IDF relationships are traditionally more useful when they are developed for a region than for a gauge station. Therefore, the city was divided into three zones using k-means clustering. Figure 7 depicts the KSNDMC stations belonging to the three zones. Zone 1 contains the fewest stations, while Zone 2 has the most. During clustering, the three groups were overlapped with the results of the ADF test. We found that the non-stationary KSNDMC stations were primarily located in Zone 3. Therefore, stationary IDF relationships were developed for Zone 1 and Zone 2 (Figure 8a,b). At the same time, a non-stationary IDF relationship was developed for Zone 3 (Figure 8c). In Figure 8, the rows represent the three return periods—10, 25, and 50 years—while the columns represent the three zones. The IDF relationship for Zone 3 yields the spatiotemporal variation in the IDF. However, the IDF relationships for Zone 1 and Zone 2 only consider a spatial variation, since the GEV parameters of the KSNDMC stations located in these zones are considered to be stationary. The results show that spatial uncertainty in the rainfall intensity decreases with the increase in the storm duration. Similarly, rainfall intensity is typically higher for low-frequency storms as compared to high-frequency ones. However, the IDF relationship between Zones 1 and 2 remains almost

the same unless non-stationarity in the KSNDMC stations is considered. In Zone 3, the 95th percentile in the location parameter results in higher rainfall intensity as compared to the 5th percentile. The results indicate that, as compared to the past, design storms may experience a higher rainfall magnitude over Zone 3 in the future. Changes in IDF relationships can be attributed to global climate and land-use changes. However, such an attribution study is beyond the scope of the present work. In conclusion, we found that spatial heterogeneity in IDF relationships exists at a finer resolution. However, at the city-region level, spatial heterogeneity is not substantial unless non-stationarity in the AMR series is considered.

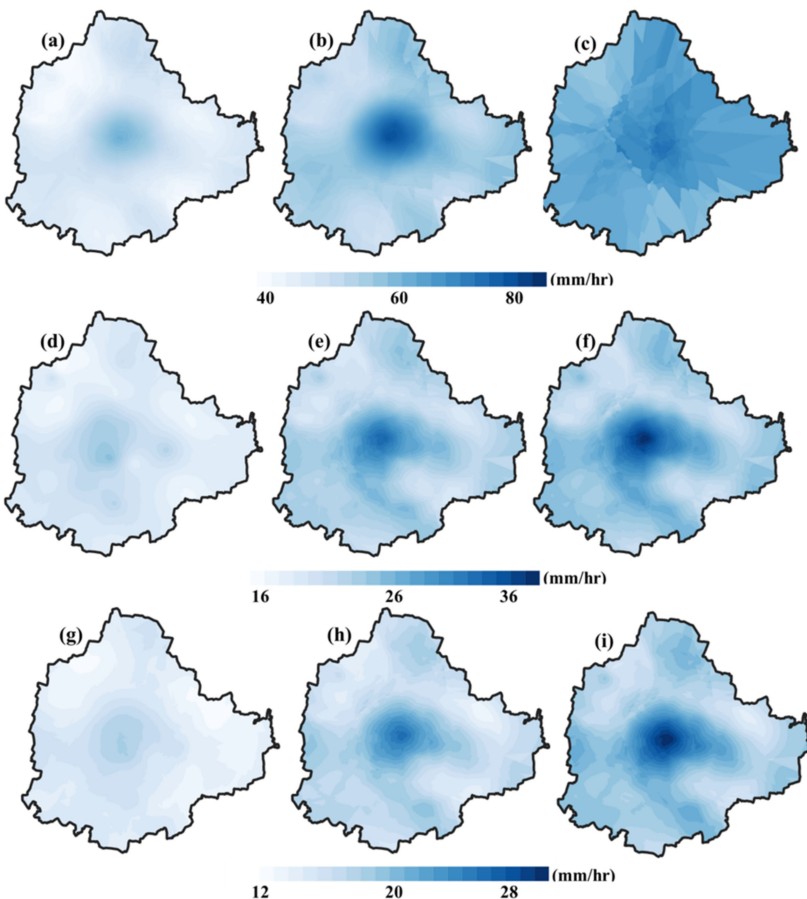

**Figure 6.** Rainfall intensity values across Bangalore for a given storm duration and frequency. Columns represent three return periods—(**a**–**c**) 10, (**d**–**f**) 25, and (**g**–**i**) 50 years—whereas the rows represents three durations: (**a**,**d**,**g**) 1, (**b**,**e**,**h**) 3, and (**c**,**f**,**i**) 5 h. KSNDMC stations that witnessed stationarity in their annual maximum hourly rainfall series were considered for the analysis. Spatial interpolation was performed according to the kriging method.

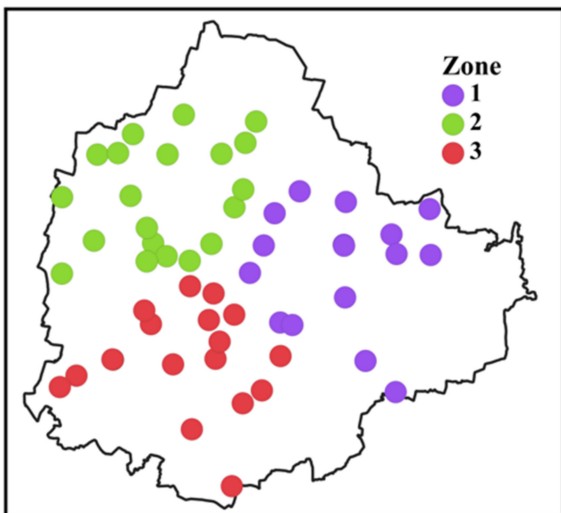

**Figure 7.** Map showing the three groups of KSNDMC stations. Clustering was performed by k-means, where grouping was carried out based on the coordinates and elevation.

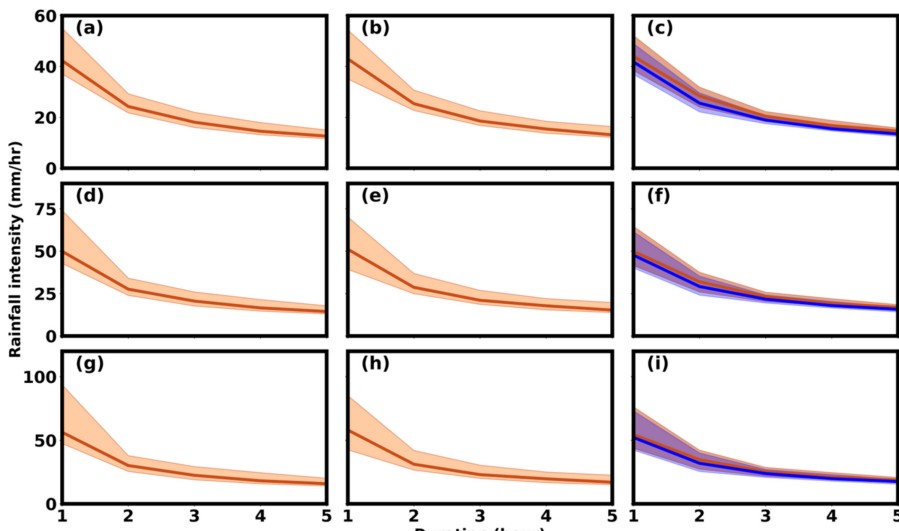

**Figure 8.** IDF relationships for the three zones across Bangalore (as the columns): (**a,d,g**) Zone 1, (**b,e,h**) Zone 2, and (**c,f,i**) Zone 3. Rows represent the three return periods: (**a–c**) 10, (**d–f**) 25, and (**g–i**) 50 years. Zone 1 and Zone 2 do not have any KSNDMC stations with non-stationarity in their annual maximum hourly rainfall series, whereas Zone 3 experiences non-stationarity in its annual maximum hourly rainfall series. For Zones 1 and 2, a spatial median, 75th percentile, and 25th percentile of the GEV parameters were considered to develop the IDF relationships, whereas for Zone 3, the 95th percentile and 5th percentile of the trend component in the location were first estimated for the KSNDMC stations. Following that, spatial processing was carried out in the same manner as in Zones 1 and 2.

### 4.3. Areal Reduction Factor

For the final task, we computed the areal reduction factor for four storm durations as a function of the circular area. The ARF values, as a function of the storm duration and circular area, are presented in Figure 9. We found that the ARF values remained close to 0.8, irrespective of the storm duration, until the circular area was less than 450 km$^2$. The ARF value for a 15-min storm was slightly lower than that for longer storms. However, as the circular area increased beyond 450 km$^2$, the ARF value decreased substantially. When the entire area was considered, the ARF value reduced to 0.4 for a 15-min storm, indicating a reduction in the areal rainfall estimate. Here, we should mention that longer storm events

were estimated from the sub-hourly (15-min duration) rainfall series. Hence, the ARF estimates related to longer storms may be subject to post-processing uncertainties. We found that the ARF values for longer storms did not decrease substantially as the circular area increased, indicating a close match between the point and areal estimates of rainfall. In summary, the ARF results indicate that an urban rain-gauge network can substantially influence urban hydraulic modeling by significantly reducing the volume of design storms.

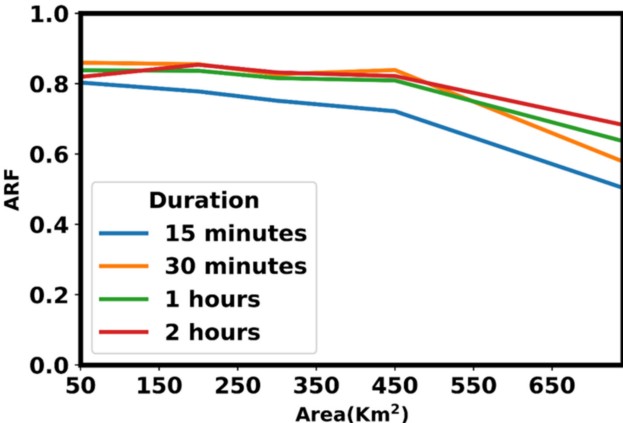

**Figure 9.** Areal reduction factor for Bangalore for different storm durations as a function of the area. The method suggested in US Weather Service Report TP-29 was used for ARF calculation.

## 5. Summary and Discussion

The present study analyzed the spatiotemporal variation in extreme rainfall for the Indian city of Bangalore. To this end, we reconstructed the historic rainfall series over a recently developed urban rain-gauge network using random forest regression. Long-term rainfall data observed from two IMD stations were considered as the predictors. Following the reconstruction, we investigated the non-stationarity in the annual maximum rainfall series of hourly reconstructed data using the ADF test. The IDF relationships were derived at the gauge level as well as at the city-region level. Finally, to understand the influence of urban rain-gauge monitoring networks over hydraulic design, we computed the areal reduction factor as a function of the storm duration and the circular area. The major findings of this study are as follows:

1.  A random forest model can efficiently reconstruct hourly rainfall time series. KSNDMC stations located near IMD stations showed higher coefficient of determination values as compared to those located farther from the IMD stations.
2.  Almost half of the KSNDMC stations exhibited non-stationarity in their AMR series, indicating that a stationary GEV model would not be sufficient to model the AMR at these stations. Additionally, non-stationarity in AMR series also implies that the IDF relationships for these stations are a function of time. We found that a non-stationary extreme value distribution with a trend component in the location parameter can efficiently model the AMR data.
3.  Substantial spatiotemporal variations exist in the IDF relationships over the KSNDMC stations and for the three city-regions. Rainfall intensity is highest at the center of Bangalore for any rainfall duration and frequency, indicating the impact of severe urbanization on the spatiotemporal characteristics of extreme rainfall. The results confirm that the IDF relationships for non-stationary grid points have been changing over the years.
4.  The ARFs for different durations are close to 0.8 until the circular area is less than 450 km$^2$. As the area increases beyond that, the ARF decreases to 0.4. The ARF results indicate that the areal average rainfall estimated from point rainfall estimates decreases as the area increases if a rain-gauge network is considered. An ARF value

between 0.4 and 0.8 indicates an overestimation in design floods if the areal average rainfall is considered directly in design flood calculation without applying the ARF.

The present study concludes that substantial variations in the extreme rainfall attributes exist within the Bangalore city limits. Furthermore, the spatiotemporal variation in the extreme rainfall attributes changes with the change in the rainfall duration and frequency. Therefore, IDF relationships vary between gauge locations within the city. Similar differences in the IDF properties are evident between city-regions as well. As the IDF relationships vary between gauge stations and city-regions, the design storm estimates within the city are also likely to vary. A similar situation was reported by a recent study that analyzed rainfall variation over Hyderabad (India) using automatic weather station records [35]. Their study found that the rainfall depth and intensity are typically higher for the oldest part of Hyderabad compared to relatively newly developed areas.

The present study could have a significant influence on urban stormwater drainage network designs. In recent years, a multi-institutional project has developed an early flood warning system for Bangalore [36]. The findings of this study could assist in the flood-warning project by improving the quality of meteorological forcing of the Storm Water Management Model (SWMM) by developing robust IDF relationships and an ARF plot for the city. Traditionally, urban drainage network designs do not consider the spatial variation in design storm estimates, which could potentially lead to the underestimation of future extreme rainfall events. This study successfully showed that a distributed urban drainage network can efficiently yield the spatiotemporal variation in extreme rainfall. However, the development and maintenance of such an urban rain-gauge network is subject to funding. As climate change and climate variability influence extreme events in the near future, it will be essential for hydraulic modelers to account for the spatial variation in design storms. The present study provides a basis to pursue additional funding for building a resilient urban stormwater drainage network.

It should be noted that there could be two major concerns related to the overall framework of this study: The first concern is related to the performance of the random forest model in efficiently reconstructing the hourly rainfall time series over the KSNDMC stations. Previous studies have found that spatially interpolated rainfall time series can be significantly different from the observed/measured series (for example, My et al., 2022 [37]). The present study found that KSNDMC stations located close to IMD stations performed better than those located farther away from the IMD stations. This is to be expected, because the local features of a region (such as the geomorphic factors, i.e., elevation, land use, lakes) influence the rainfall at the KSNDMC stations. Hence, the association between IMD rainfall and KSNDMC rainfall weakens as the distance between the two stations increases. Therefore, the reconstructed hourly rainfall series for the historic period might not provide an appropriate representation of the observed rainfall. In turn, an inappropriate reconstruction of the hourly rainfall series could potentially impact the estimation of IDF relationships. The present study took multiple steps to ensure an appropriate representation of the hourly rainfall series, including (i) considering seasonality in the model cross-validation, (ii) preprocessing the model inputs, and (iii) employing multiple machine learning models (although the results are not shown). Alternatively, hierarchical models (such as Bayesian dynamic hierarchical models) could be considered to account for the spatial correlation among the KSNDMC stations. A similar analysis cannot be efficiently performed with the original KSNDMC station data for the period 2010–2019, since this would lead to large uncertainties in the IDF relationships resulting from the low sampling variability in the annual maximum rainfall series. Additionally, this study assumes that the functional relationships between the predictand and predictor series remain stationary over the calibration/validation and reconstruction periods. This assumption does not restrict the model in transferring the non-stationarity in the IMD rainfall time series to the KSNDMC rainfall time series. Previous studies have shown that linear models cannot yield the non-stationarity in predictand time series if it is absent in the predictand series [38,39]. The present study found that almost half of the stations had experienced non-stationary

in the reconstructed time series; therefore, a stationary functional relationship between the predictand and the predictor does not play an influential role in the current problem. Furthermore, a 'stationary' assumption is very common for bias correction and statistical downscaling models for future projections [40]. It should be noted that the reconstruction of urban hydrometeorological observations is a challenging task, since efficient reconstruction approaches (such as tree-ring chronology) cannot be applied in the absence of input data. The second major concern regards the non-stationary extreme value distribution (EVD). This study assumes that non-stationarity in the EVD may arise in the location parameter. Furthermore, a linear trend in the location parameter was considered in this study. However, it should be noted that non-stationarity in the EVD may arise in the shape parameters, in the scale parameters, or in any combination of parameters. Such non-stationarity may not necessarily follow a linear trend. In such cases, the EVD parameters and hyperparameters can be estimated from a maximum likelihood estimation. The best non-stationary EVD can be selected based on the performance criteria. However, the present study refrained from considering non-stationarity in the shape and scale parameters because the linear trend in the location parameter was statistically significant.

The spatiotemporal variation in extreme rainfall within the city can be primarily attributed to geomorphic features and population changes. Additionally, hydro-climatological variables—such as SST teleconnection—can also potentially influence the spatiotemporal variation, as they have a strong association with urban rainfall [41,42]. Climate change and climate variability could have enhanced the spatiotemporal variation in cities as they emerged as global hotspots. Bangalore, in particular, has spatial variations in its elevation and has experienced a significant increase in population over the last few decades; both could have potentially influenced the statistical attributes related to the extremes. It should be noted that the identification of the drivers influencing the spatiotemporal variability is beyond the scope of this study. However, a future study could relate the potential predictors (such as land use and land cover, population, elevation, etc.) with the EVD parameters and hyperparameters using a simple multivariate linear regression model. Several cities around the globe have experienced a similar situation to that of Bangalore, i.e., an increase in population along with the growing impact of climate change. Therefore, spatiotemporal variations in the extreme rainfall characteristics can be expected in other such cities as well. If an urban rain-gauge network is developed, it would be possible for researchers to estimate a robust areal reduction factor that could assist in hydraulic design. For Bangalore, this study found a significant decrease in the ARF with the increase in the circular area, indicating a substantial reduction in the volume of the design storms. In conclusion, this study demonstrated the presence of spatiotemporal variations in the extreme rainfall attributes of the city, which should be addressed with the assistance of an urban rain-gauge network.

**Supplementary Materials:** The following supporting information can be downloaded at: https://www.mdpi.com/article/10.3390/w14233900/s1, Figure S1: Location of the city of Bangalore. Figure S2: Histogram showing the number of stations, with respective *p*-values; 27 stations clearly have stationary rainfall time series, while the others are non-stationary. Figure S3: Elbow curve, where k = 3 is chosen. Table S1: Correlation between model-predicted extremes and observed extremes for 30 KSNDMC stations. Extreme rainfall is estimated using a point-over-threshold approach; the 95th percentile of sub-daily rainfall is taken as the threshold. Table S2: Number of stations in each zone.

**Author Contributions:** Conceptualization, R.D.B., R.J. and P.P.M.; software, validation, investigation, R.J.; writing, R.D.B. and R.J.; supervision, P.P.M.; writing—review and editing, P.P.M. and R.D.B. All authors have read and agreed to the published version of the manuscript.

**Funding:** The study was primarily funded by the project "Urban Modelling: Development of Multi-sectorial Simulation Lab and Science-Based Decision Support Framework to Address Urban Environment Issues" (Sanction Number MeitY/R&D/HPC/2(1)/2014) from the Ministry of Electronics and Information Technology (MeITy), under the National Supercomputing Mission program of Government of India; PI-Prof PP Mujumdar. The study was also partially funded by INSPIRE faculty

fellowship (IFA18-ENG262) from the Department of Science and Technology, Government of India, awarded to Dr. R Das Bhowmik.

**Data Availability Statement:** The data presented in this study are available upon request from the corresponding author. The IMD and KSNDMC data can be obtained from the respective organizations. The IMD data can be found here: https://dsp.imdpune.gov.in/ (accessed on 1 January 2021). The KSNDMC data can be found here: https://www.ksndmc.org/ReportHomePage.aspx (accessed on 1 January 2021).

**Acknowledgments:** The authors would like to thank the India Meteorological Department, Pune, and the Karnataka State Natural Disaster Monitoring Centre for providing the necessary data.

**Conflicts of Interest:** The authors declare no conflict of interest. The funders had no role in the design of the study; in the collection, analyses, or interpretation of data; in the writing of the manuscript; or in the decision to publish the results.

## Appendix A. The Random Forest Regression Algorithm

Random forest regression is considered to be one of the most efficient general-purpose learning techniques [43]. It is primarily a collection of decision trees, where the outcome is obtained as the average of predictions from all of the trees. Decision trees are made of nodes, where each node is treated as a junction that splits the data based on certain conditions. In the present study, the squared error of sample data was considered as the function to measure the quality of the split. The major disadvantage of using single decision trees is that they overfit the training data, leading to low predictive accuracy. The problem of overfitting is avoided by building many individual trees, and the same procedure is extended and represented as a random forest [29]. The present study considered 10 decision trees.

Compared to decision trees, a random forest algorithm calculates error rates with a higher accuracy. In a random forest, as the number of trees increases, the error rate converges. The out-of-bag (OOB) error is computed using the training set, which gives a good estimate of the performance of the forest on unseen data. Hence, separate model training and model validation are not required—the modeler can utilize the entire dataset.

## Appendix B. Augmented Dickey–Fuller Test

The augmented Dickey–Fuller test (ADF test) is a statistical significance test used to investigate whether a given time series is stationary or non-stationary. The null hypothesis states that a unit root ($\alpha$ in Equation (A1)) is present and, hence, the time series is non-stationary. The mathematical expression for the Dickey–Fuller test is provided below.

$$y_t = c + \beta t + \alpha y_{t-1} + \phi \ \Delta Y_{t-1} + e_t \tag{A1}$$

The Dickey–Fuller test investigates whether the unit root in Equation (A1) is 1. In Equation (A1), $y_{t-1}$ is the lag1 of the time series, while $\Delta Y_{t-1}$ is the first-order difference of the time series. The augmented Dickey–Fuller test evolved based on Equation (A1), which includes higher-order regressive terms and is one of the most common forms of unit root test.

$$y_t = c + \beta t + \alpha y_{t-1} + \phi_1 \Delta Y_{t-1} + \phi_2 \Delta Y_{t-2} + \ldots \ \phi_p \Delta Y_{t-p} + e_t \tag{A2}$$

Since the null hypothesis assumes the presence of a unit root, the statistical significance of the unit root can also be estimated as a $p$-value, which should be less than the significance level (e.g., 0.05) in order to reject the null hypothesis. A histogram exhibiting $p$-values across the KSNDMC stations is presented in the Supplementary Materials (Figure S2)

## Appendix C. K-Means Clustering

K-means clustering is a commonly used unsupervised learning algorithm to group a set of data. The grouping of data, known as clustering, is performed to achieve minimal coherence within the clusters. Coherence is measured as sum of squares, and it is termed

as inertia. The optimal number of clusters is chosen from an elbow curve, as shown in the Figure S3 the Supplementary Materials. An elbow curve is a plot of the within-cluster sum of squares (wcss), which is a measure of the variability of the observations within each cluster and the number of groups, $k$ (in the present study, $k$ varies from 1 to 5).

$$wcss = \sum_{l=1}^{k} \sum_{p=1}^{m} distance(d_p, c_l) \tag{A3}$$

where $c$ is the cluster centroid, while $d$ represents the data points in each cluster. The number of clusters varies from 1 to $k$, and the number of data points within the cluster varies from 1 to $m$. In the present study, latitude, longitude, and elevation were chosen as the parameters to decide how to group the station data. Table S2 in the Supplementary Materials shows the number of stations in each zone.

### Appendix D. Areal Reduction Factor

The areal reduction factor (*ARF*) relates the point rainfall to the areal rainfall. Urban hydraulic modeling requires the knowledge of temporal and spatial variability of average rainfall over a region, making *ARF* analysis essential for the modeling. The *ARF* values range from 0 to 1. The present study used the following equation, originally suggested by the Technical Paper 29 (U.S. Weather Bureau 1957), to calculate the *ARF*:

$$ARF_{TP-29} = \frac{\frac{1}{n}\sum_{j=1}^{n}\hat{R}_J}{\frac{1}{k}\sum_{i=1}^{k}\left(\frac{1}{k}\sum_{j=1}^{n}R_{ij}\right)} \tag{A4}$$

where

$k$ = number of stations in the area.
$n$ = number of years.
$R_{ij}$ = annual maximum point rainfall for year $j$ at station $i$ = max $(r_{i1}^{j}, r_{i2}^{j} \ldots r_{id}^{j})$.
$d$ = number of specific durations in the year.
$r_{iu}^{j}$ = specific duration of point precipitation at station $i$ in year $j$ on day $u$.
$R_j$ = annual maximum areal rainfall for year $j$ = max $(\hat{r}_1^{j}, \hat{r}_2^{j} \ldots \hat{r}_d^{j})$
$\hat{r}_1^{j}$ = specific duration of areal precipitation at specific time $u$ for year $j$.

The equation for ARF was obtained from the work of Allen and Degaetano, 2005. In this study, durations of 15 min, 30 min, 1 h, and 2 h were considered for the areas ranging from 100 km$^2$ to 740 km$^2$. The circular area was calculated using the procedure suggested by Allen and Degaetano in 2005 [34].

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
