# Peer review of "Reconstruction of Urban Rainfall Measurements to Estimate the Spatiotemporal Variability of Extreme Rainfall"

_water, doi:10.3390/w14233900_

Round 1

Reviewer 1 Report

The authors use a machine learning approach to reconstruct observations from a higher-density precipitation observing network in Bangalore with a relative short observing period (after 2010) using data from IMD rainfall stations that have been operating since 1969. This is an interesting approach to addressing the sparsity of rainfall observations in the earlier part of the record, but I have a significant concern about the approach  and feel the manuscript should be returned to the authors for Major Revisions prior to publication to see if it can be addressed.

Stationarity of Derived Relationship: The authors use the period of overlap between the IMD and KSNDMC stations, 2010-2019, to train their random forest method. This relationship is then used to reconstruct the values at the KSNDMC stations going back to the beginning of the IMD data in 1969. However, this approach assumes that the relationship derived for the 2010-2019 is valid for the entirety of the record, i.e. that the relationship between the IMD and KSNDMC data is stationary. It is not obvious to me that this is a valid assumption, and the authors make no mention its possible effect on their conclusions. Since on of the major goals of the manuscript is to identify spatio-temporal changes in rainfall within the city, ignoring possible changes in the relationship areas housing the IMD and KSNDMC stations in time seems like a serious deficiency. 

Author Response

Response: The authors agree to the reviewer’s concern that stationarity in the functional relationship between predictand and predictor stations has been assumed. However, the assumption does not restrict the model to transfer the nonstationarity in IMD rainfall timeseries (the predictor) to the KSNDMC rainfall time series (the predictand)--analysis of which is one of the primary objectives of the current study. Machine learning models can not yield the nonstationarity in predictand time series if the nonstationarity is absent in the predictand series (see Seo et al., 2019, Das Bhowmik et al., 2019). The current study has shown that almost half of the stations have experienced nonstationarity in the reconstructed time series; therefore, a stationary functional relationship between the predictand and the predictor does not play an influential role in the current problem. Finally, a ‘stationary’ assumption is very common for bias-correction and statistical downscaling models for future projections (for example, Mehrotra and Sharma, 2012).

The discussion section has been revised to reflect these points [L454-468]

Bhowmik, R. D., & Sankarasubramanian, A. (2019). Limitations of Univariate Statistical Downscaling to Preserve Cross-Correlation between monthly precipitation and temperature. International Journal of Climatology. 1-18. https://doi.org/10.1002/joc.6086.

Seo, S., Bhowmik, R. D., Sankarasubramanian, A., Mahinthakumar, G., & Kumar, M. (2019). The role of cross-correlation between precipitation and temperature on basin-scale simulation of hydrologic variables. Journal of Hydrology. Journal of Hydrology, 570, 304-314.

Mehrotra, R., & Sharma, A. (2012). An improved standardization procedure to remove systematic low frequency variability biases in GCM simulations. Water Resources Research, 48(12).

Reviewer 2 Report

The paper summarizes the results of a study aimed to estimate the spatial variability of rainfall intensity distribution over the city of Bangalore, based on time series reconstructed using Random Forest Regression.

Here are a few comments:
- in the abstract, you write that "The study applies forest-tree regression", which sounds a bit weird. Did you maybe mean "tree-based regression"?

- the word coefficient is misspelt (co-efficient) once in subchapter 3.1 and twice in subchapter 3.1. Please fix it. 

- Figure 3: since the score is a continuous variable, why didn't you just use a graduated color bar instead of dividing the data into 3 classes? There is a significant loss of information in this case, especially in class 0.6 - 1, which comprehends strongly different values. To a reader, this would appear unnecessarily hideous and must definitely be fixed.

-  Figure 4 is not very visible, maybe a scatter plot reconstructed Vs. observed would be more effective, even if it doesn't show the temporal relationship. Maybe also a scatter plot of the time series could be fine. Please consider these alternative visualizations.

- Figures 5b and 5c: since you are using the same y-axis for parameters with very different meanings and ranges, it is almost impossible to understand the distribution of shape and beta1. Please consider using also a right y-axis for shape and beta1, or even deleting 5b and 5c and adding a table with the 5th, 25th, 50th, 75th, and 95th percentiles of the values.

- In the caption of Figure S2 you write that the histogram "Clearly indicates stations with stationary rainfall time series are of 27 in count". In my opinion, it is not necessary to comment on the results in the caption. Furthermore, since the top of the bar is closer to 30 than 25, it is clearly not 27, but rather 29 or 28. Please consider removing it or checking the data.

The idea behind the paper is interesting, but with the provided information is impossible to reproduce your work and draw conclusions from it, since there are no details about the regression model. How did you tune the hyperparameters of the random forest?
Which data did you use as input to obtain the value at time t? Were they just the IMD stations values at time t? Please provide a detailed description.

Figure 6 is in my opinion the most important figure in the paper since it summarizes the spatial variation of the rainfall intensity. Since it also includes 50 yrs return periods, I guess that the whole reconstructed data series have been used for this.

However, spatially-interpolated rainfall time series can be significantly different from the measured series. You could cite this paper, which already discusses this issue:
My, L.; Di Bacco, M.; Scorzini, A.R. On the Use of Gridded Data Products for Trend Assessment and Aridity Classification in a Mediterranean Context: The Case of the Apulia Region. Water 202214, 2203. https://doi.org/10.3390/w14142203

Given that in your case the spatial distribution of the rainfall intensity resembles the spatial distribution of the reconstruction accuracy, I suggest simple further analysis to support your conclusions and dissipate doubts.

Even if the original KSNDMC series are too short to obtain high return period values, you could still use it to further investigate the spatial distribution. Please consider comparing the spatial distribution of the top 0.1% rainfall intensity values of the original KSNDMC (and maybe also other percentiles, if you want) with the spatial distribution of Figure 6 to support your conclusions.

Author Response

Reviewer 2

Comments and Suggestions for Authors

The paper summarizes the results of a study aimed to estimate the spatial variability of rainfall intensity distribution over the city of Bangalore, based on time series reconstructed using Random Forest Regression.

Here are a few comments:

- in the abstract, you write that "The study applies forest-tree regression", which sounds a bit weird. Did you maybe mean "tree-based regression"?

Response: Changed to ‘Random forest regression’.

- the word coefficient is misspelt (co-efficient) once in subchapter 3.1 and twice in subchapter 3.1. Please fix it.

Response: This is fixed in the revised manuscript.

- Figure 3: since the score is a continuous variable, why didn't you just use a graduated color bar instead of dividing the data into 3 classes? There is a significant loss of information in this case, especially in class 0.6 - 1, which comprehends strongly different values. To a reader, this would appear unnecessarily hideous and must definitely be fixed.

Response: Thanks for the suggestion. However, following a sincere consideration, the authors did not change the current figure as the discreet ranges shown in the figure align with Devore’s (2011) suggestion.  Devore (2011) suggested that a correlation value higher than 0.8 is considered to be “strong,” while a correlation value between 0.5 and 0.8 is regarded as “moderate.” Also, replacing the discreet ranges with continuous color ranges would affect the visual appearance of the plot as the output is not at grid points but across KSNDMC stations.

Devore, J. L. (2011). Probability and statistics for engineering and the sciences. Boston, USA: Cengage learning.

-  Figure 4 is not very visible, maybe a scatter plot reconstructed Vs. observed would be more effective, even if it doesn't show the temporal relationship. Maybe also a scatter plot of the time series could be fine. Please consider these alternative visualizations.

Response: The figure has been revised. Thanks for the valuable suggestion

- Figures 5b and 5c: since you are using the same y-axis for parameters with very different meanings and ranges, it is almost impossible to understand the distribution of shape and beta1. Please consider using also a right y-axis for shape and beta1, or even deleting 5b and 5c and adding a table with the 5th, 25th, 50th, 75th, and 95th percentiles of the values.

Response: The figure has been revised with a secondary Y-axis. Thanks for the valuable suggestion.

- In the caption of Figure S2 you write that the histogram "Clearly indicates stations with stationary rainfall time series are of 27 in count". In my opinion, it is not necessary to comment on the results in the caption. Furthermore, since the top of the bar is closer to 30 than 25, it is clearly not 27, but rather 29 or 28. Please consider removing it or checking the data.

Response: The authors agree with the reviewer. The comment has been deleted.

The idea behind the paper is interesting, but with the provided information is impossible to reproduce your work and draw conclusions from it, since there are no details about the regression model. How did you tune the hyperparameters of the random forest?

Response:  The study has followed rigorous model calibration and validation to ensure no data-overfitting. Additionally, the same model can be run multiple times to develop a model parametric uncertainty which is not very high in the current case (results not shown). A manual trial and error procedure was carried to decide the number of trees in the algorithm, which can vary from station to station. Additional remarks are included in subsection 3.1 (L165-169).

Which data did you use as input to obtain the value at time t? Were they just the IMD stations values at time t? Please provide a detailed description.

Response: This has already been mentioned at the end of subsection 3.1. The predictor of the model at any given time is the observations at two IMD stations. Hence, the predictors are not varying for KSNDMC stations; only the functional form between the prediction and predictors varies between KSNDMC stations. We have revised subsection 3.1 to clarify the reconstruction (L177-179).

Figure 6 is in my opinion the most important figure in the paper since it summarizes the spatial variation of the rainfall intensity. Since it also includes 50 yrs return periods, I guess that the whole reconstructed data series have been used for this.

However, spatially-interpolated rainfall time series can be significantly different from the measured series. You could cite this paper, which already discusses this issue:

My, L.; Di Bacco, M.; Scorzini, A.R. On the Use of Gridded Data Products for Trend Assessment and Aridity Classification in a Mediterranean Context: The Case of the Apulia Region. Water 2022, 14, 2203. https://doi.org/10.3390/w14142203

Response: The entire reconstructed time series has been to perform IDF analysis. This is done to increase the sampling variability of the annual maximum rainfall.

Thanks for suggesting the article. We have cited the article in the revised manuscript. (L440)

Given that in your case the spatial distribution of the rainfall intensity resembles the spatial distribution of the reconstruction accuracy, I suggest simple further analysis to support your conclusions and dissipate doubts.Even if the original KSNDMC series are too short to obtain high return period values, you could still use it to further investigate the spatial distribution. Please consider comparing the spatial distribution of the top 0.1% rainfall intensity values of the original KSNDMC (and maybe also other percentiles, if you want) with the spatial distribution of Figure 6 to support your conclusions.

Response: The authors thank the reviewer for this valuable suggestion. However, carrying out the same analysis but with original KSNDMCl data will lead to a huge amount of uncertainty in IDF since the 2010-2019 years of the original dataset results in only ten observations for annual maximum rainfall (if block maxima are considered). However, if the POT (as top 0.1% rainfall intensity) is considered, then sampling variability of AMR might be appropriate, but we have to consider the generalized Pareto distribution, which will cause a mismatch with reconstructed IDF since reconstructed IDF is carried out considering a GEV distribution. This is an important comment; we have discussed it in the revised article.

The discussion section has been revised to reflect these points [L454-468]

Round 2

Reviewer 2 Report

I'm glad to see that the paper has been slightly improved and the typos have been removed, so it is suitable for publication in my opinion.

That said, I recommend adjusting figure 3; I still think a graduated colour bar would be the more suitable choice for the coefficient of determination, which is a scalar value. However, if you really want to use classes, I suggest using the class limits suggested by Devore et al.

Author Response

Dear Reviewer,

A revised figure is provided as per your suggestion.

Sincerely